# The Effect of Different Plant Oil Impregnation and Hardening Temperatures on Physical-Mechanical Properties of Modified Biocomposite Boards Made of Hemp Shives and Corn Starch

**DOI:** 10.3390/ma13225275

**Published:** 2020-11-21

**Authors:** Dovilė Vasiliauskienė, Giedrius Balčiūnas, Renata Boris, Agnė Kairytė, Arūnas Kremensas, Jaunius Urbonavičius

**Affiliations:** 1Department of Chemistry and Bioengineering, Vilnius Gediminas Technical University, LT-10223 Vilnius, Lithuania; dovile.vasiliauskiene@vgtu.lt; 2Institute of Building Materials, Faculty of Civil Engineering, Vilnius Gediminas Technical University, LT-08217 Vilnius, Lithuania; giedrius.balciunas@vgtu.lt (G.B.); renata.boris@vgtu.lt (R.B.); agne.kairyte@vgtu.lt (A.K.); arunas.kremensas@vgtu.lt (A.K.)

**Keywords:** oil-impregnation, biocomposite board, plant oils, physical-mechanical properties, hemp shive aggregate, corn starch binder

## Abstract

In this study, tung tree and linseed drying oils, as well as semi-drying hempseed oil, were analyzed as the protective coatings for biocomposite boards (BcB) made of hemp shives, corn starch binder, and the performance-enhancing additives. The hydrophobization coatings were formed at 40, 90, and 120 °C temperatures, respectively. The physical-mechanical properties such as the compressive strength, thermal conductivity, dimensional stability, water absorption, and swelling were tested. In addition, scanning electron microscopy (SEM) was employed for the analysis of the board microstructure to visualize the oil fills and impregnation in pores and voids. It was demonstrated that the compressive strength of oil-modified BcBs compared to uncoated BcBs (at 10% of relative deformation) increased by up to 4.5-fold and could reach up to 14 MPa, water absorption decreased up to 4-fold (from 1.34 to 0.37 kg/m^2^), swelling decreased up to 48% (from 8.20% to 4.26%), whereas the thermal conductivity remained unchanged with the thermal conductivity coefficient of around 0.085 W/m·K. Significant performance-enhancing properties were obtained due to the formation of a protective oil film when the tung tree oil was used.

## 1. Introduction

The increased interest in plant resources as renewable materials is growing in recent years. The biological origin of raw ingredients allows the production of environmentally friendly materials, but biodestructive factors emerge during the exploitation. While using this kind of materials, e.g., thermal insulation boards, moisture could accumulate at the partition walls of the building due to the hygroscopicity and lead to favorable conditions for the growth of microorganisms [1].

Previous studies have reported that hemp shives absorb a large amount of water due to the hydroxy groups present in their constituents such as cellulose, hemicellulose, and lignin. Another important factor contributing to water sorptivity of hemp shives and their-based materials is the porosity [2].

The ability to absorb large amounts of moisture promotes the growth of microorganisms, which results in the breakdown of plant cell walls and less durability of materials. To reduce water sorption into bio-based materials, the hydrophilicity needs to be converted into the hydrophobicity [3].

Impregnation with vegetable oils is the classical method of preservation of bio-based materials. Semi-drying and drying oils are a perfect way of forming hydrophobic coatings. The hydrophobic coating with the vegetable oils protects materials not only from the water but also from the destructive effects of microorganisms, termites, and nematodes, and also is the protection against UV radiation [4]. Therefore, modification processes through chemical and thermal methods are effectively applied to oils of plant origin involved in coating formulations [4].

Drying oils have received more attention compared to other oils because of crosslinking to a tough, solid film after exposure in air, even at room temperature, through autoxidation without evaporation of water or other solvents. Tung tree oil (TO) is one of the most abundant drying oils whose major component is a triglyceride of an α-eleostearic ((9Z,11E,13E)-octadeca-9,11,13-trienoic) acid. Compared to other unconjugated drying oils, such as soybean and linseed oils, tung oil has a faster drying time, better water resistance, and greater hardness. Tung oil has a highly unsaturated and conjugated triene structure, therefore it can easily be chemically modified [5]. Linseed oil obtained from the seeds of Linum usitatissimum penetrates the surface of the bio-based materials and dries by an autoxidation process [6]. While linseed oil provides highly appreciated optics and haptics to wood surfaces, protection against water through surface hydrophilization and the one against mechanical wear are provided only minimally [6]. Hempseed oil was described in several studies as a useful product in pharmaceutical, nutrition, and cosmetic industries [7,8]. Hempseed oil is known as a bio-based coating for materials, but it is proposed to modify this type of oil before performing the hydrophobic coating [9].

The above-described oils are largely composed of triglyceride mixtures. The autoxidation mechanism increases the viscosity of the oils, which hardens over time until stable C-C bonds are formed. During this process, the esters formed between one molecule of glycerol and three molecules of various linear fatty acids. The most common fatty acids encountered in drying oils are the saturated acids with 12, 14, 16, or 18 carbon atoms (lauric, myristic, palmitic, and stearic acids, respectively) and C18 polyunsaturated acids with 1, 2, or 3 double bonds (oleic, linoleic, and linolenic acids) (Table 1) [10].

Previously, several groups have investigated the oils as the hydrophobic coatings either on the wood or wood shive boards [15,16]. In addition, hemp shives themselves were coated by sol-gel coatings to increase the hydrophobicity [3]. Here, we propose a new ecological way to protect from the moisture the biocomposite boards (BcBs) made from the hemp shives by forming the hydrophobic coatings from natural raw oils. To achieve this, the biocomposite boards were formed and coated by different natural oils, such as either the hempseed, linseed, or tung tree. The mechanical and thermal characteristics of such boards were measured. It was demonstrated that the protective coatings increase the compressive strength and decrease the water absorption and swelling of BcBs.

## 2. Materials and Methods 

### 2.1. Preparation of BcBs and Their Impregnation with Oils of the Plant Origin

Biocomposite boards were prepared from hemp shives, purchased from local farmers in Rokiškis region of Lithuania with a particle size varying from 2.5 to 5 mm. As a binder, the corn starch with a bulk density of 550 kg/m^3^ was purchased from the “Roquette”, Lestrem, France. As the flame retardants, either an expandable graphite (G additive) NORD-MIN 249, “Nordmann, Rassmann GmbH”, Hamburg, Germany or a multifunctional aqueous mixture based on phosphorus and nitrogen organic compounds (F additive) “Flovan CGN-01” obtained from “Hunstmann” Basel, Switzerland, were used.

Composition of BcB forming mixtures was as follows:

H—hemp shives with 10% corn starch as the binder (composition H).

G—hemp shives with 10% corn starch and expandable graphite (20% of the starch mass, composition G).

F—hemp shives with the 10% corn starch and the 30 g/L concentration flame retardant aqueous mixture “Flovan” (composition F).

Biocomposite boards were formed from the prepared mixtures by pressing them to 40% of their initial volume at a pressure of 0.8 MPa (the same pressure for all specimens was applied). Then, the loaded specimens were fixed with screws and taken for the thermal treatment. The thermal treatment of biocomposite boards consisted of the following stages: raise of temperature (within 1 h, from the room temperature to 160 °C), maintenance of temperature (160 °C temperature is maintained for 6 h in the thermal treatment chamber), and the decrease of temperature (within 3 h, till the room temperature). In total, 18 boards (6 for each composition) at (400 × 400 × d) mm^3^ were formed.

For the formation of the protective coating, semi-drying hempseed oil was purchased from the farmers in Rokiškis region of Lithuania. Its density was estimated at 0.924 ± 0.002 g/L. The drying linseed oil was obtained from the farmers in Utena region of Lithuania. Its density was estimated at 0.929 ± 0.002 g/L. The drying tung tree oil was purchased from UAB “Pro colore”, Vilnius Lithuania. Its density was estimated at 0.941 ± 0.002 g/L.

For the impregnation, the biocomposite boards were covered with the selected oils amounting at about 43–94% total weight of BcB, and exposed in the vacuum desiccator about 0.9–1 bar pressure for 5 min. This procedure was repeated for 5–10 times. After this procedure, BcBs with oil were maintained at either 40, 90, 120 °C temperature and 1% humidity conditions in the thermostat (22 °C was used as control) for 24–240 h or for the time required to reach the stable weight.

### 2.2. Oil Thin Film Micro-Oxidation Test

A 500 μm layer of test oil was formed on the polished steel disc of 19 mm diameter. The roll with oil was heated at 90 °C temperature in an electric convection oven (Snoll 120/60), for 24–240 h. Samples were tested at least in triplicate, removing the data from runs with the visible oil spills. In every 2 h during the first 1 h period, the disc was cooled down to the room temperature for measurement of the weight loss (gain). Oxidation duration was recorded as the aggregate in the first 24 h with the 2 h time intervals, and then after 24, 72, 240 h during which the rolls were placed in the oven. The change in mass is calculated according to Equation (1). The time spent on cooling and weighing was not included:(1)$$\Delta m(\%)=100\;-\left(\left(\frac{m_{g}-m_{0}\;\;}{m_{i}-m_{0}}\right)\times \;100\right),$$
where *m_g_*—disc mass after 90 °C heating, *m*_0_—disc mass without oil, *m_i_*—disc and oil mass before the heating.

### 2.3. Measurements of Physical and Mechanical Properties

Thermal conductivity of hemp shives was determined following EN 12667 standard [17]. The test was carried out using a FOX 304 (LaserComp, New Castle, DE, USA) computerized testing machine, which has the measurement limits from 0.01 to 0.50 W/(m·K) and measuring accuracy of ~1%. The difference between measuring plates was 20 °C and the average test temperature was 10 °C. For the tests, three samples of each composition were formed. Before the measurements, test samples were dried in a ventilated drying oven at 70 °C to a constant mass (the difference between two weightings made in 24 h period was <0.1%) and then conditioned to a constant mass for not less than 72 h to equilibrium at a temperature of (23 ± 5) °C and relative air humidity of (50 ± 5)%. To determine moisture content gained during the measurement, specimens were weighed before and after the thermal conductivity determination. The percentage of open pores was determined according to EN ISO 4590 standard [18] method 2 for (100 × 30 × 30) mm^3^-sized BcB specimens.

Compressive stress at 10% of deformation was tested according to the EN 826 standard [19] using a computerized machine H10KS (Hounsfield, Surrey, UK) with a maximum loading force of 10 kN, a loading accuracy of ±0.5%, and a loading speed accuracy of ±0.05%. Three specimens for each composition with a size of (50 × 50 × d) mm^3^ (d–thickness of specimen) were used. Before the test, specimens were conditioned for not less than 6 h at (23 ± 5) °C and relative air humidity of (50 ± 5)%. Then, the specimen was aligned onto the bottom support and loaded with an initial loading of (250 ± 10) Pa. The loading speed during the tests was (0.1∙d ± 25%) mm/min and the specimen was compressed until 10% of deformation.

Water absorption by partial immersion was determined according to EN ISO 29767 standard [20] method A. Five specimens with a size of (50 × 50 × d) mm^3^ were used. Before the test, specimens were conditioned for not less than 6 h at (23 ± 5) °C and relative air humidity of (50 ± 5)%.

Determinations of swelling in thickness after water immersion was performed according to the EN 317 standard [21]. Five specimens of (50 × 50 × d) mm^3^ size were used, after conditioning at (20 ± 2) °C ambient temperature and (65 ± 5)% relative humidity. All specimens were immersed in a water bath at (20 ± 1) °C, pH (7 ± 1).

Determination of dimensional stability under specified temperature and humidity conditions were determined according to EN 1604 [22] at 70 °C and 90% RH environmental conditions. Three specimens with a size of (200 × 200 × d) mm^3^ were used. Before the test, specimens were conditioned at (23 ± 2) °C temperature and (50 ± 5)% relative humidity for 14 days until the changes in linear dimensions were <0.1%.

The microstructure of the samples was investigated using a JSM-7600F field emission scanning electron microscope (SEM) (JEOL, Tokyo, Japan). The parameters of the SEM tests were as follows: voltage 4 and 10 kV; distance to specimen surface 9–11 mm; magnification 1000× and 1500×. Before testing, the specimens were covered with an electrically conductive thin Au layer by evaporating an Au electrode in vacuum using a Quorum Q150R ES instrument (Quorum, Laughton, UK).

To perform a statistical interpretation of the results, Statsoft software tool was implemented. Linear and non-linear correlations were used for the determination of the relationship between variable. The proper description model for the mathematical relationship was determined after comparing determination coefficients of a few possible models. To evaluate the scattering of experimental results on both sides of the regression curve, the average standard deviations, $S_{r}$ were calculated. Regardless of predicted dot values, ${\bar{Y}}_{x_{i}}$ the possible error values δ, that enabled interval prediction, were calculated by Equation (2):(2)$$Y_{x_{i}}^{pred.}={\bar{Y}}_{x_{i}}\pm \delta ,$$
while the value of possible errors can be calculated using Equation (3):(3)$$\delta =t_{\alpha }\cdot S_{r},$$
where $t_{\alpha }$—Student’s criterion that is selected based on the degree of freedom, f=m−n when the confidence interval is 95%.

## 3. Results and Discussion

### 3.1. Oxidation of Different Types of Oils and Formation of Oil Films

A micro-oxidation test was performed to determine the oil hardening time. For the formation of the protective coating, 90 °C temperature was used. This temperature stimulates the autoxidation of oil, as shown as previously [10]. The autoxidation increased viscosity of the oil, which hardens over the time; this occurs according to the polycondensation mechanism, low molecular weight compounds are released, it gradually becomes less flowable, more viscous, and loses solubility.

The protective coating using the hempseed oil was formed after 240, but not after 24 h (Figure 1a,b); the complete hardening was observed after 96 h, as determined by the stabilization of mass. This is a longer period comparing with the formation of coatings observed when either linseed or the tung oil was used. This is probably due to the natural antioxidants as tocopherols and polyphenols present in a raw hempseed oil [23]. The linseed oil has fully hardened in 78 h. During the first 24 h, the linseed oil (Figure 1c) was at the fast stage of the cross-link formation [24]. After that, it turned into the thin film, with a gel structure. Then, the hardening process was slowly continuing, until the hard coating was formed and the mass did not change (Figure 1d). The fastest formation of the hard coating was observed when the tung tree oil was used (Figure 1e). A hard, porous yellowish coating was formed already after 2 h (Figure 1e). After 24 h, the color turned into the dark yellow, and after 240 h—into the light orange-brown. This phenomenon is observed probably due to the presence of α-eleostearic acid (Table 1).

Molecular labile cross-links give rise to highly stable networks that are the basis for the hydrophobic and protective coating. Autoxidation of the unsaturated fatty acids is assumed to increase the oil mass. The incubation for 240 h at 90 °C led to a different level of the autoxidation and thereby the weight increase (Table 2). As mentioned above, the slowest autoxidation was observed for the hempseed oil due to the presence of the antioxidants such as vitamin E [23].

The early stages of oxidation of triglycerides, corresponding to the drying phase, consist of the autoxidation phenomenon of the unsaturated fatty acid components that occurs with the formation of conjugated unsaturated links and with the development of extensive cross-linking. On the following stage, the slow consumption of the labile cross-links gives rise to a highly stable network, which still contains unreacted triglycerides and low molecular weight molecules formed by the fragmentation [10].

### 3.2. Oil Penetration into the Biocomposites Boards

Oil penetration depends on the BcB composition (Table 3). The total accessible porosity of hemp shives is 76.67 ± 2.03% [2].

The F type of BcB with multifunctional flame retardants absorbed the hempseed oil to 55.05%, linseed oil—to 63.41%, and tung tree oil—to 43.33%. A multifunctional aqueous mixture based on phosphorus and nitrogen organic compounds (Flovan) was previously used for textile [25]. Such additive increases the hydrophobicity, thereby reducing the penetration of the oil into the BcB.

The penetration of all three oil types into G type of BcB was about 20% higher (about 82–93% in total) than for the other types of BcBs. Expanded graphite is used as the adsorbent for crude oil, diesel, and lubricant oil. In the first 12 h, the expanded graphite adsorbs the lubricant oil at 4.5 g for 1 g of graphite [26]. That observation may explain the oleophilic properties of expanded graphite and high level of penetration into the BcB boards.

In the H type of BcB without any additives, the oil penetrated 61.64% to 70.23%, which was consistent with the hypothesis that all empty pores would be filled with the oil.

### 3.3. Water and Moisture Resistance Properties of Oil-Impregnated BcBs

Hemicellulose and amorphous cellulose are responsible for the high water absorption of plant-based materials and composites since they contain numerous easily accessible hydroxyl groups which give a hydrophilic character to such aggregates [27]. The possibility of using BcBs in outdoor applications makes it necessary to analyze their water uptake behavior. Figure 2, Figure 3 and Figure 4 represent the results obtained while measuring respectively the short-term water absorption by partial immersion, swelling in thickness and dimensional stability under increased temperature and humidity conditions of BcBs, impregnated by different oils.

Maximum water absorption was obtained for BcBs that were oil-impregnated and hardened at 40 °C temperature (Figure 2). It is worth mentioning that non-impregnated control BcBs have the average value of water absorption equal to 4.5 kg/m^2^ for F composition, 4.2 kg/m^2^ for G composition and 4.7 kg/m^2^ for H composition. Tung tree oil showed the highest efficiency and, compared to control BcBs, reduced water absorption by 36% for F-type, by 50% for G-type, and 45% for H-type boards. These results show that hardening of oil at 40 °C is only slightly effective. In contrary, when the oils were hardening at either 90 or 120 °C, much less water absorption was observed. Irrespective of the BcB board type (F, G, or H), hempseed and linseed oils had almost the same impact; water absorption values were reduced more than 2-fold for the boards prepared at 90 °C compared to the ones prepared at 40 °C temperature and more than 3-fold compared to control BcBs without the oil-impregnation. Besides, it was shown that the most effective hardening temperature is 120 °C at which the action of different oil types depends on the composition of BcBs, i.e., BcBs with F and G compositions had the lowest water absorption with tung tree oil impregnation—1.0 kg/m^2^ and 0.90 kg/m^2^, respectively, while for BcBs with H composition it was for linseed oil—0.37 kg/m^2^. Independently on the impregnation temperature, the biggest reduction of water absorptivity was observed when the tung tree oil was used. This behavior can be a result of tung tree oil hydrophobicity and structure, which forms a thick layer on the external surface as well as porous aggregate structure. Similar conclusions were made previously for rice husk and soy protein-based particle boards impregnated with tung tree oil [28,29]. However, H-type BcB has increased water absorption, which, compared to the results obtained for F-type and G-type BcBs, suggests that F-or G-additives together with the tung tree oil form some kind of synergy, i.e., F-additive forms a thin protective layer on hemp shive surface, whereas G-additive interferes with corn starch binder.

As it was observed previously [30], due to hydrophobic nature, additives such as G weaken the ability of corn starch to bind water when performing the water absorption test, thus creating the effective paths for water to enter. However, tung tree oil effectively penetrates those cracks or voids formed during incorporation of additives, hereby reducing water absorption of BcBs.

To additionally evaluate the scattering of the results of water absorption for all BcBs impregnated with different oils, statistical calculations and interpretations of the results were implemented. Statistical data of short-term water absorption by partial immersion results are presented in Table 4.

Analysis performed shows that the obtained results of short-term water absorption can be approximated by the regression equations. Table 5 presents the constant coefficients $b_{0}$, $b_{1}$, $b_{2}$, the average standard deviations, and the determination coefficients. The obtained data show that the determination coefficients are nearly 1, thus suggesting that the proposed regression models are suitable to describe the water absorption values of different oils-impregnated BcBs which were hardened at the temperature interval of (40–120) °C.

To further evaluate the water penetration into BcBs, swelling in thickness experiments were conducted. As for water absorption, similar changes in swelling in thickness results observed and are summarized in Figure 3. It can be observed that contrary to water absorption, the results of swelling in thickness changes in a polynomial manner.

Independently of the oil type and hardening temperature used, swelling in thickness was reduced compared to the untreated BcBs. It was observed that 40 °C hardening temperature was not effective, i.e., compared to control BcBs, all oil types increased swelling in thickness values up to 1.5-fold for all compositions. A slight positive impact could be observed at 90 °C hardening temperature when either hempseed or linseed oil was used. As demonstrated previously for water absorption, the use of such oils provides similar effects for BcBs. Improved results were obtained for tung tree oil-impregnated BcBs at hardening temperature of 90 °C. Compared to control BcBs, it reduced swelling in thickness by ~15.7% for F composition, by ~48.0% for G composition, and by 10.4% for H composition. Higher impact of the impregnation with tung tree oil was observed likely due to its capability to form a more cross-linked and dense film on the surface of BcB constituents. Stronger bonds between the raw materials do not allow corn starch particles to swell, thus restricting not only water absorption and swelling in thickness but also the volumetric changes. Similar conclusions were done for cellulose and starch composites crosslinked with the citric acid that blocked some of the hydroxyl groups and increased the water resistance [31]. The same conclusions could be done for 120 °C hardening temperature. Hempseed and linseed oil influenced similarly all compositions, i.e., slightly reduced swelling in thickness, but due to variation of measurement data, no significant impact was observed. However, the tung tree oil significantly reduced the swelling in thickness, from 8.6% to 4.6% for F composition, from 8.2% to 5.8% for G composition, and from 8.9% to 6.5% for H compositions, i.e., up to 2-fold.

To additionally evaluate the scattering of the results of swelling in thickness for all BcBs impregnated with different oils, mathematical-statistical calculations and interpretations of the results were implemented. Statistical data are presented in Table 5. Research shows that the obtained results of short-term water absorption can be approximated by the regression equations. Table 5 presents the constant coefficients $b_{0}$, $b_{1}$, $b_{2}$, the average standard deviations $S_{r}$, and the determination coefficients $R_{T\cdot \Delta S_{t}}^{2}$.

The obtained data show that the determination coefficients are nearly 1, thus suggesting that the proposed regression models are suitable to describe the swelling in thickness values of different oils-impregnated BcBs which were hardened at the temperature interval of (40–120) °C.

The degree in which plant-based composites shrink and/or swell when moisture content changes is an important property determining its suitability for different applications. This parameter is known as dimensional stability. The targeted dimensional changes are not standardized for BcBs, but acceptable ones according to EN 1604 are 2.5% for products tested under increased temperature and humidity conditions. The results of dimensional stability in thickness are presented in Figure 4.

It is seen that hardening at 40 °C is inefficient; the improvement of the dimensional stability can be observed only at 90 and 120 °C temperatures. The dimensional stability of BcBs prepared with oils at hardening temperature of 90 °C in all cases improved by up to 3-fold and did not exceed the limit of 2.5%, indicating an effective impregnation. As shown previously [29], oil impregnation reduces the number of hydroxy groups of plant-based composites, thus increasing dimensional stability of all BcBs. The highest improvement at 90 °C hardening temperature was observed for H-type compositions, leading to the conclusion that the oils freely penetrate the pores, voids, and cracks formed between corn starch binder and hemp shives, thus hindering the changes in thickness. However, the F-additive is distributed throughout hemp shive surface, and oil penetrates only via cracks and voids existing in corn starch binder as can be seen from oil penetration results in Table 3. Therefore, the F additive-corn starch interface might be not strong enough to withstand the impact of moisture.

In addition, G-additive acts as an insert interfering with the contact zones between corn starch binder and hemp shives, thereby leading to somewhat worse dimensional stability compared to H-composition. Some interesting changes can be observed when the oil-hardening temperature is increased to 120 °C. It can be hypothesized that at a higher temperature before the oil gets solid, the viscosity is reduced, allowing better penetration into the BcB structure (Figure 5) but reducing the thickness of the final film which is not capable of withstanding the swelling power. Dimensional stability at increased temperature and humidity conditions, also the water absorption and swelling in thickness results show that the tested oils (hempseed, linseed, and tung tree) are capable of increasing the hydrophobicity of the corn starch. These observations are in full agreement with previous studies [32,33].

### 3.4. Mechanical Performance of Oil-Impregnated BcBs

Strength characteristics of the BcBs are impacted by the oil type and heat treatment due to heat-induced alteration of BcBs chemical structure of valve and vessels wall components. It is demonstrated that a high oil uptake also contributes to the higher mechanical performance of BcBs compared to the other heat treatment methods [34]. Therefore, it is not necessary to use high temperatures to efficiently impregnate wood-based or similar products. Figure 6 demonstrates the compressive stress at 10% deformation results versus hardening temperature of three types oil-impregnated BcBs with or without the additives.

Additionally, the obvious improvement of the compression strength is seen at 40 °C oil-hardening temperature, where it increased 1.5–3-fold compared to control BcBs, suggesting that oil as a wood impregnator activates itself at 40 °C. Similar improvement was observed previously [35] where Poplar and Robinia wood were impregnated for 2–6 h at 160 °C temperature with sunflower, linseed, and rapeseed oils. The difference in activation temperatures might be attributed to the different impregnation duration times and oil types.

The first strength rise can be noticed at 90 °C oil-hardening temperature where it reaches its highest values: 7.3 MPa for hempseed oil, 8.1 MPa for linseed oil, and 9.3 MPa for tung tree oil in F compositions; 8.9 MPa for hempseed oil, 9.4 MPa for linseed oil, and 14 MPa for tung tree oil in G compositions; and 6.5 MPa for hempseed oil, 7.9 MPa for linseed oil, and 11 MPa for tung tree oil in H compositions (Figure 6). It is assumed that the vegetable oils fill vessels and valves and thickens the walls, cross-links, forms a denser film, and owes to better lateral stability to the hemp shives, which normally fail in compression due to buckling of thin valves and vessels walls. Similar findings were reported by other researchers who investigated the dimensional stability and mechanical behavior of the oil-impregnated wood and particleboards [28,36]. Increasing the hardening temperature up to 120 °C shows the negligible change in strength value for F composition, while it becomes similar to those at 40 °C oil-hardening temperature for G and H composition.

The obtained results demonstrate that strength parameters increased most when the tung tree oil was used for hardening at 90 °C. A significant improvement of the compressive stress at 10% deformation suggest a synergistic effect of oil used for the impregnation and starch used as a binding material for the production of BcBs. The synergy may be explained by the hydrogen bonding between the free carboxylic groups from the hemp, linseed, and tung tree oils molecules with the hydroxy groups in the corn starch [37].

### 3.5. Density and Thermal Conductivity of Oil-Impregnated BcBs

The above-described results demonstrated that the most appropriate performance is obtained for BcBs prepared with linseed or tung tree oil and at 90 °C hardening temperature. Therefore, further analysis was made for BcBs prepared with such oils and hardening conditions. In most cases, density plays an important role in the performance characteristics of BcBs; Figure 7 represents how these characteristics vary in different BcBs prepared using different oil types and hardened at 90 °C.

Regardless of the oil type used for impregnation, density increase was observed in all cases. The highest increase in density was seen for dry specimens that were impregnated with tung tree oil (48%) whereas the lowest one—for air-dried specimens impregnated with linseed oil (37%). This significant increase is observed due to the impregnation technology, which allows oils to freely penetrate structure and fill the pores, voids in BcBs and the valves, vessels, tracheids in shives themselves (Figure 8). The successful oil-impregnation and hardening of wood-based products at higher temperature intervals was confirmed by several previous studies [18,26,27]. Moreover, the density changes in BcBs with linseed oil-impregnation and BcBs with tung tree oil-impregnation is due to the difference in fatty acids of impregnation oils.

Tung tree oil has averagely a higher number of carbon-carbon double bonds, therefore, it dries more rapidly and cross-links more densely thus forming a thicker film in BcBs structure which is, according to microstructural analysis, ~1 μm, while for linseed oil it is ~0.6 μm. Additionally, the difference in densities between H, F, and G compositions is negligible. The increment may be explained by the additional usage of expandable graphite which increases the density [38,39].

To use the prepared BcBs for the thermal insulation, the conductivity measurements were done. Figure 9 presents the values of the differently prepared biocomposite boards with different oils and dried at different conditions (the same as used for density measurements in Figure 7).

The highest increment of thermal conductivity (34%) was obtained for dry BcB samples that were impregnated with tung tree oil, whereas the lowest one (22%)—for air-dried linseed oil-impregnated BcBs. Similar thermal conductivity values we observed previously [40] in studies on biocomposites with various pine-based biochar obtained at (550–600) °C temperature and impregnated with coconut oil. After the oil-impregnation, the increase in thermal conductivity values directly correlates with the increase of density of the BcBs (Figure 7). As stated before, during the impregnation, the oil penetrates the porous structure of BcBs and forms a highly cross-linked and dense film on the interface of BcBs constituents. For this reason, the area of contact zones increases, thus determining the heat loss through conduction. To evaluate the efficacy of oils to fill in or isolate the pores, open porosity determination was performed and is presented in Table 6. As demonstrated there, oil-impregnation minimizes open porosity, thus locking the pathways for the penetration of water molecules and assuring better interaction between raw materials. The most efficient type of oil that reduces the porosity is tung tree oil which, irrespective of the composition used, averagely reduces open porosity and voids by ~40%, whereas the linseed oil-by ~30% compared to control BcBs.

In addition, as plant-based materials are hygroscopic [41], non-impregnated BcBs have a thermal conductivity which increases up to 4.3% at air-dry conditions. While, thermal conductivity between the oil-impregnated oven-dry and air-dry BcBs do not change or change within the measurement error of the apparatus. Therefore, it may be assumed that selected BcB treatment technologies allow minimizing the hygroscopicity of BcBs.

Furthermore, as in the case of density, the thermal conductivity of G-composition was observed to be higher than that of H- and F-compositions. It is because the expandable graphite has a crystalline structure and is an excellent heat conductor with phonons that are the main heat carriers [42,43] compared to the other raw materials used for the preparation of the BcBs. Therefore, upon adding the expandable graphite up to 20% (by starch weight) to the BcB matrix, thermal conductivity slightly increases for all BcBs independently on the conditioning method, i.e., from 0.0639 W/(m∙K) to 0.0667 W/(m∙K) for dried to constant weight samples and from 0.0667 W/(m∙K) to 0.0697 W/(m∙K) for air-dried samples. The obtained results are consistent with the other studies reporting that graphene– or graphite–polymer interfaces have high thermal conductivity due to phonon scattering [44,45].

## 4. Conclusions

In this work, biocomposite boards were formed and impregnated using the oils of plant origin. Due to the presence of C=C bonds in such oils, the protective films were formed by the autooxidation and polycondensation processes. It was demonstrated that such coating increases the compressive strength and decreases the water absorption and swelling of BcBs. Tung tree oil suited most of the oils tested for these purposes increasing the compressive strength by about 4.5-fold and reduction of swelling in thickness by 48% for the BcB G-composites that contained the expandable graphite and when oil coating was formed at 90 °C. For such kind of BcBs, the water absorption was decreased by about 2.8-fold, whereas the thermal conductivity slightly increased. SEM observations supported these conclusions, showing the excellent penetration oils into BcB structure to fill the pores and voids. Regardless of the oil type used for impregnation, the density of the BcBs increased in all cases.

The obtained BcBs, especially the ones with H and G compositions, can find their application in the furniture industry due to their even and decorative surface as well as sufficient mechanical and moisture resistance performance. Additionally, the raw materials used are not hazardous to human health and are environment friendly. Therefore, the obtained BcBs can be a perspective alternative to the existing ones, e.g., wood particleboards with formaldehyde, epoxides, or isocyanates. However, further extensive research is necessary to determine the optimal technological parameters, allowing the reduction of time required for production and impregnation of BcBs.

## Figures and Tables

**Figure 1 materials-13-05275-f001:**
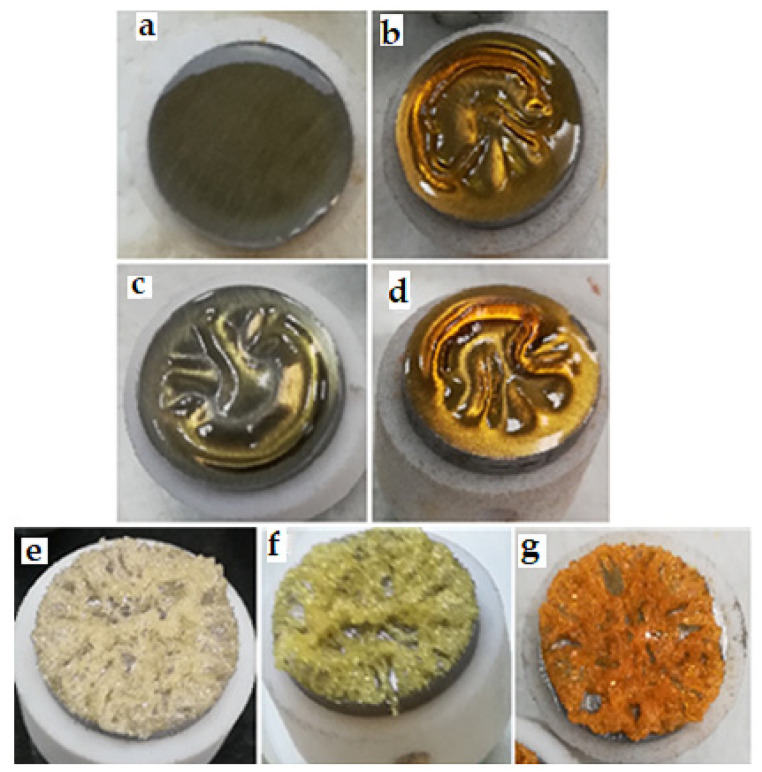
The spicemens after incubation at 90 °C temperature: hempseed oil—24 h (**a**), 240 h (**b**); linseed oil—24 h (**c**), 240 h (**d**); tung tree oil 2 h (**e**), 24 h (**f**), 240 h (**g**).

**Figure 2 materials-13-05275-f002:**
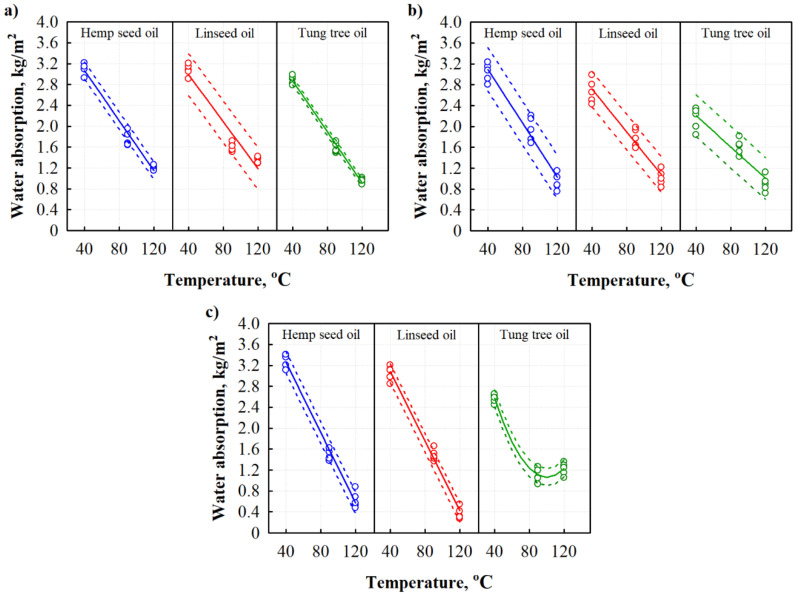
Dependences of water absorption on hardening temperature of different oils-impregnated BcBs: (**a**) F composition; (**b**) G composition; and (**c**) H composition.

**Figure 3 materials-13-05275-f003:**
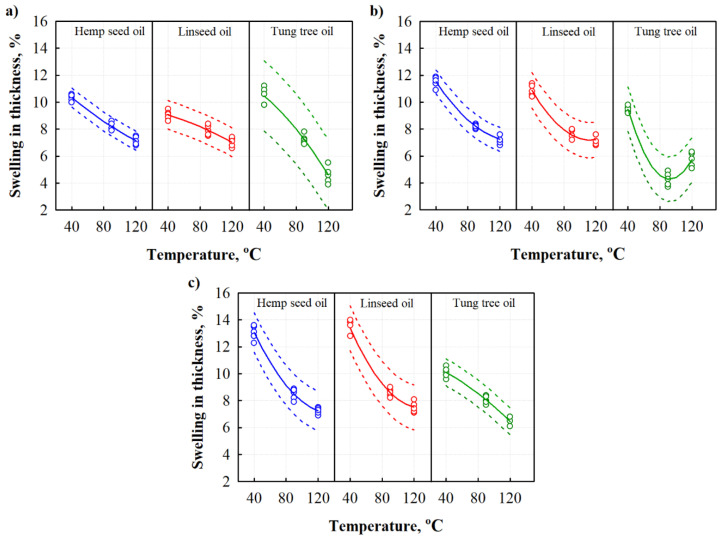
Dependences of swelling in thickness on hardening temperature of different oils-impregnated BcBs: (**a**) F composition; (**b**) G composition; and (**c**) H composition.

**Figure 4 materials-13-05275-f004:**
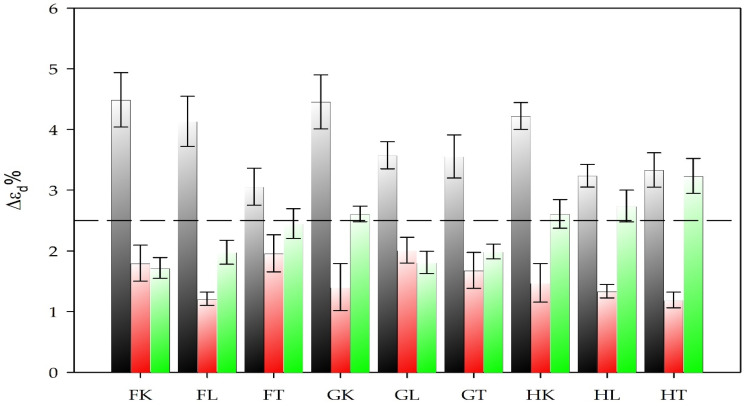
Dimensional stability of different oil-impregnated BcBs at hardening temperatures: 
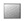
 40 °C, 
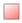
 90 °C, 
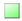
 120 °C. F, G and H—BcB composition, K—hempseed oil, L—linseed oil, T—tung tree oil, and Δεd—the change in thickness.

**Figure 5 materials-13-05275-f005:**
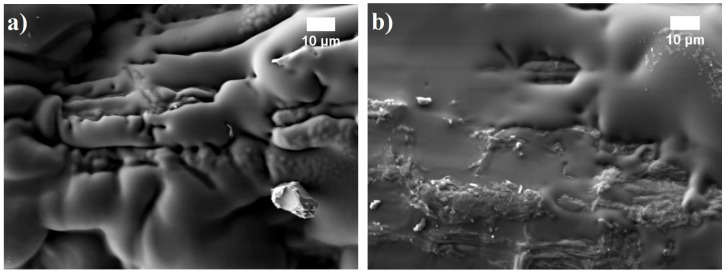
Solid tung tree oil on hemp shives aggregate at hardening temperature: (**a**) 90 °C and (**b**) 120 °C (magnification 1000×).

**Figure 6 materials-13-05275-f006:**
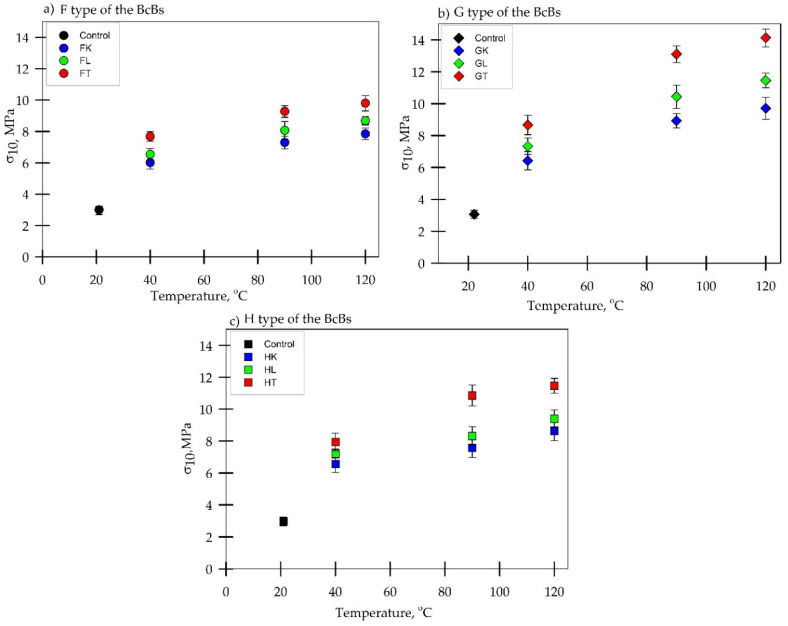
Compressive stress at 10% deformation of BcBs with different oil-impregnation at different temperatures:(**a**) F type of the BcB with hempseed oil (K), linseed oil (L), and tung tree oil (T); (**b**) G type of the BcB with hempseed oil (K), linseed oil (L), and tung tree oil (T); (**c**) H type of the BcB with hempseed oil (K), linseed (L), and tung tree oil (T).

**Figure 7 materials-13-05275-f007:**
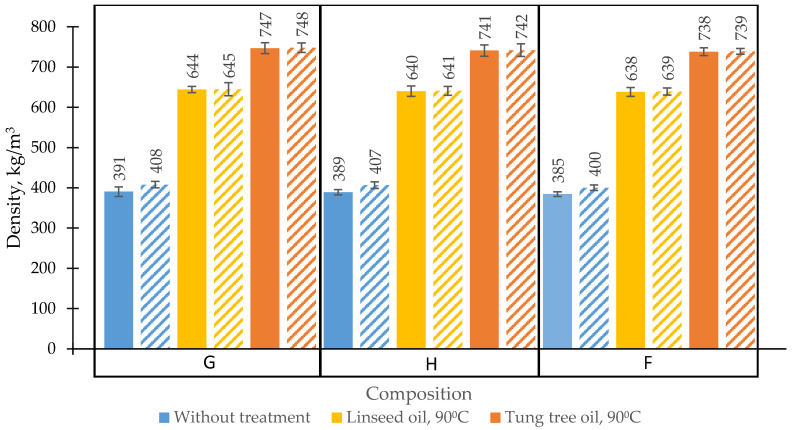
The density of different oil-impregnated BcBs. G, H, F—composition of the boards, impregnated at 90 °C hardening temperature. 
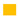
—linseed oil, 
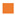
—tung tree oil, 
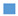
—without oil treatment: solid-type bars—specimens which were dried at 70 °C to constant weight; pattern-type bars—specimens which were dried at 23 ± 2 °C and 50 ± 2% conditions to constant weight.

**Figure 8 materials-13-05275-f008:**
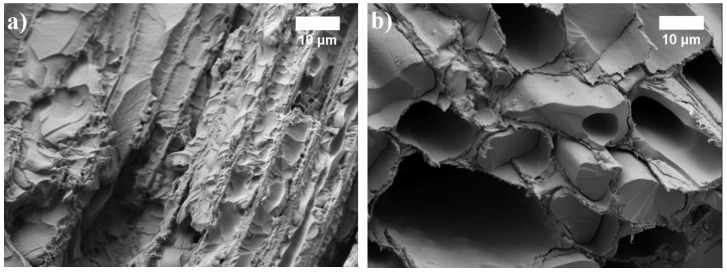
The microstructure of BcBs with different oil-impregnation and hardening at 90 °C: (**a**) linseed oil; (**b**) tung tree oil (magnification ×1500).

**Figure 9 materials-13-05275-f009:**
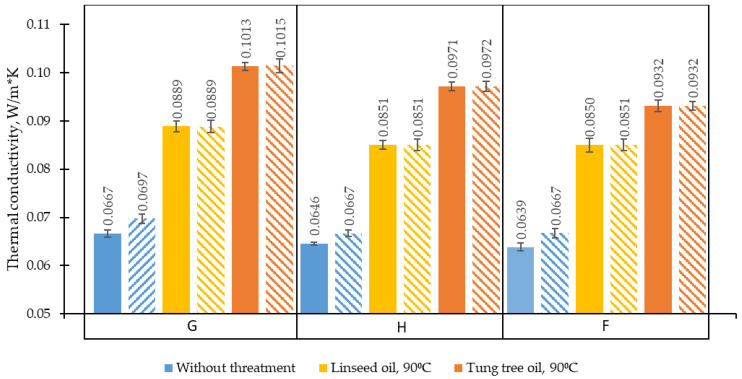
Thermal conductivity of different oil-impregnated BcBs. G, H, F—composition of boards, impregnated at 90 °C hardening temperature. 
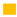
—linseed oil, 
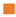
—tung tree oil, 
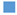
—without oil treatment: solid-type bars—specimens which were dried at 70 °C to constant weight; pattern-type bars—specimens which were conditioned at (23 ± 2) °C and (50 ± 2)% to constant weight.

**Table 1 materials-13-05275-t001:** Composition of the drying oils.

Fatty Acid Composition (%)	Tung Tree Oil [5,11]	Linseed [12,13,14]	Hempseed Oil [8,14]
Palmitic acid (C16:0)	3–5.2	6–7	4–8
Oleic acid (C18:1)	4–11	14–24	8–17
Stearic acid (18:0)	2	3–6	2–3
Linoleic acid (C18:2)	7.5–15	14–19	50–70
α-linolenic acid (C18:3(n-3))	3	48–60	12–25
γ-linolenic acid (C18:3(n-6))	nd	nd	0–6.8
α-eleostearic acid (C18:3(9Z,11E,13E))	59–83.3	nd	nd

nd—not determined.

**Table 2 materials-13-05275-t002:** The total mass changes due to the autooxidation of different oil types.

Oil	Total Changes % of the Mass
Hempseed	1.77 ± 0.52
Linseed	2.87 ± 0.18
Tung tree	3.49 ± 0.34

**Table 3 materials-13-05275-t003:** Oil penetration data for different oil-impregnated biocomposite boards (BcBs) with various compositions.

Type of the BcB	Characteristic	Hempseed Oil	Linseed Oil	Tung Tree Oil
F	Oil penetration into the BcB, %	55.1 ± 12.6	63.4 ± 24.5	43.3 ± 5.81
G	Oil penetration into the BcB, %	81.8 ± 14.2	93.4 ± 20.6	85.5 ± 19.7
H	Oil penetration into the BcB, %	61.6 ± 16.9	68.2 ± 14.6	70.2 ± 18.2

**Table 4 materials-13-05275-t004:** Statistical data of short-term water absorption by partial immersion results.

Type of Composition ^(1)^	Equation Coefficients			$R_{T\cdot W_{sp}}^{2}$	$S_{r}$, kg/m^2^
	$b_{0}$	$b_{1}$	$b_{2}$		
FK *	4.028878	−0.024027	–	0.976	0.0645
FL *	3.887878	−0.022527	–	0.936	0.156
FT *	3.841122	−0.024373	–	0.989	0.0298
GK *	4.127531	−0.025818	–	0.949	0.163
GL *	3.535184	−0.020510	–	0.935	0.132
GT *	2.805347	−0.015008	–	0.867	0.156
HK *	4.603347	−0.033608	–	0.985	0.0799
HL *	4.411469	−0.033282	–	0.988	0.0601
HT **	5.251000	−0.083675	0.000418	0.977	0.0432

* Formula—$W_{sp}=b_{0}+b_{1}\cdot T$; ** Formula—$W_{sp}=b_{0}+b_{1}\cdot T+b_{2}\cdot T^{2}$, where $W_{sp}$—short-term water absorption by partial immersion, kg/m^2^; $b_{0}$, $b_{1}$, $b_{2}$—constant coefficients which were calculated from the experimental data; T—hardening temperature, °C. ***NOTE:** Two-sided prediction confidence interval of the results is presented with 95% probability, Student’s criterion*
$t_{\alpha }=2.751$
*when*
α=0.05
*(dotted lines in Figure 2)*. ^(1)^ FK, FL and FT–BcBs with Flovan additive and hemp seed, linseed and tung tree oils, respectively; GK, GL and GT–BcBs with expandable graphite additive and hemp seed, linseed and tung tree oils, respectively; HK, HL and HT–control BcBs without additives and with hemp see, linseed and tung tree oils, respectively.

**Table 5 materials-13-05275-t005:** Statistical data of swelling in thickness results.

Type of Composition ^(1)^	Equation Coefficients			$R_{T\cdot \Delta S_{t}}^{2}$	$S_{r}$, %
	$b_{0}$	$b_{1}$	$b_{2}$		
FK	12.45200	−0.057067	0.000107	0.964	0.275
FL	9.678000	−0.012183	−0.000082	0.873	0.414
FT	11.97400	−0.026050	−0.000295	0.960	1.01
GK	15.72400	−0.122300	0.000430	0.976	0.344
GL	15.98400	−0.154800	0.000680	0.957	0.510
GT	20.51000	−0.352083	0.001908	0.969	0.646
HK	18.98600	−0.173283	0.000628	0.978	0.575
HL	19.91000	−0.192417	0.000742	0.977	0.649
HT	11.16800	−0.020433	−0.000157	0.960	0.389

Formula—$\Delta S_{t}=b_{0}+b_{1}\cdot T+b_{2}\cdot T^{2}$, where $\Delta S_{t}$—swelling in thickness, %; $b_{0}$, $b_{1}$, $b_{2}$—constant coefficients which were calculated from the experimental data; T—hardening temperature, °C. ***NOTE:** Two-sided prediction confidence interval of the results is presented with 95% probability, Student’s criterion*
$t_{\alpha }=2.751$
*when*
α=0.05
*(dotted lines in Figure 3)*. ^(1)^ FK, FL and FT–BcBs with Flovan additive and hemp seed, linseed and tung tree oils, respectively; GK, GL and GT–BcBs with expandable graphite additive and hemp seed, linseed and tung tree oils, respectively; HK, HL and HT–control BcBs without additives and with hemp see, linseed and tung tree oils, respectively.

**Table 6 materials-13-05275-t006:** Open porosity of different oil-impregnated BcBs at 90 °C temperature.

Parameter	Type of BcB		
Open Porosity, vol.%	**G**	H	F
	Control BcB		
	18.3 ± 0.77	20.2 ± 0.59	18.1 ± 0.57
	BcB with linseed oil		
	12.2 ± 0.64	13.7 ± 0.80	12.5 ± 0.62
	BcB with tung tree oil		
	10.1 ± 0.64	11.3 ± 0.69	11.0 ± 0.46

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
