# Peer review of "The Effect of Different Plant Oil Impregnation and Hardening Temperatures on Physical-Mechanical Properties of Modified Biocomposite Boards Made of Hemp Shives and Corn Starch"

_materials, 2020, doi:10.3390/ma13225275_

Round 1

Reviewer 1 Report

The manuscript is within the scope of Materials Journal and is generally well written. However, before publication some concerns must be addressed, namely:

1 - Words such as "best" should be avoided since are not very accurate.

2 - The plots from Figure 7 must be improved.

3 - Please delete the horizontal lines from the plots in Figures 8 and 10 and use thick in the left axis of the plot indicating the values (no need to prolong the lines).

4 - In Figure 9, please put the a) on the left side and the b) figure on the right side.

5 - Conclusions - avoid the use of "best" use instead showed improved behaviour when compared to...(for instance)

Author Response

Thank you for the valuable remarks on our manuscript. Our responses are presented point-by-point below.

Point 1: Words such as "best" should be avoided since are not very accurate.

Response 1: The sentence in an abstract was modified and now stands like this: "Significant performance-enhancing properties were obtained due to the formation of protective oil film when the tung tree oil was used."

The sentence "The obtained results demonstrate that best strength parameters are obtained when the tung tree oil was used for hardening at 90ᴼC." was modified into "The obtained results demonstrate that strength parameters increased most when the tung tree oil was used for hardening at 90ᴼC."

Point 2: The plots from Figure 7 must be improved.

Response 2: Figure 7 was redrawn and improved, now stands as Figure 6.

Point 3: Please delete the horizontal lines from the plots in Figures 8 and 10 and use thick in the left axis of the plot indicating the values (no need to prolong the lines). 

Response 3: Figures 8 and 10 were modified and now stand as Figures 7 and 9.

Point 4: In Figure 9, please put the a) on the left side and the b) figure on the right side.

Response 4: Done; figure 9 now stands as Figure 8.

Point 5: Conclusions - avoid the use of "best" use instead showed improved behaviour when compared to...(for instance).

Response 5: The sentence "Tung tree oil was the best one of the oils tested for these purposes increasing the compressive strength by about 4.5-fold and reduction of swelling in thickness by 48% for the BcB G-composites that contained the expandable graphite and when oil coating was formed at 90oC." was modified into "Tung tree oil suited most of the oils tested for these purposes increasing the compressive strength by about 4.5-fold and reduction of swelling in thickness by 48% for the BcB G-composites that contained the expandable graphite and when oil coating was formed at 90oC." 

Reviewer 2 Report

In this study, tung tree and linseed drying oils as well as semi drying hempseed oil were analysed as the protective coatings for biocomposite boards. Nowadays the topic is in current interests for the readers. Anyway, authors should made changes and additions in the text.

Keywords:

  • Authors should change the term „physical and mechanical properties“ into one „property“

Introduction

  • Authors have correctly presented the state of the art.

Materials and Methods

  • The density values oft he oils in the text should be given (mainly the range from min. up to max. value). The table 2 is not necessary.
  • Figure 1 is not necessary.
  • Informations about amount of samples and conditioning parameters in the laboratory should be given (e.g. temperature, relative humidity).

Results and Discussion

  • The time period required for the formation of protective coating in the text should be given. The table 3 is not necessary.
  • There is a lack of the statistical estimation of the results from the tables 4-5 and figures 3-5. It is necessary for the scientific paper.

Conclusions

  • Authors wrote „…Biocomposite boards made of the materials of natural origin are sensible to the environmental factors. In order to increase their longevity and improve the physical-chemical and mechanical properties, impregnation of boards was made using the oils of plant origin...“. These are not conclusions. Authors should delete these sentences.

I recommend the paper for the publishing after minor changes.

Author Response

Thank you for the valuable remarks on our manuscript. Our responses are presented point-by-point below.

Point 1: Authors should change the term „physical and mechanical properties“ into one „property“.

Response 1: This term was changed into "physical-mechanical" properties.

Point 2: The density values oft he oils in the text should be given (mainly the range from min. up to max. value). The table 2 is not necessary.

Response 2: Table 2 was deleted, and the values were given in the text.

Point 3: Figure 1 is not necessary.

Response 3: This figure was removed.

Point 4: Informations about amount of samples and conditioning parameters in the laboratory should be given (e.g. temperature, relative humidity).

Response 4: This information is given in sections 2.1 and 2.3. The sentence "Before the test, specimens were conditioned for not less than 6 h at (23±5)°C" was amended with "...and relative air humidity of (50±5)%." 

Point 5: The time period required for the formation of protective coating in the text should be given. The table 3 is not necessary.

Response 5: Table 3 was deleted and the times needed to form protective oil coatings were mentioned in the text.

Point 6: There is a lack of the statistical estimation of the results from the tables 4-5 and figures 3-5. It is necessary for the scientific paper.

Response 6: The upper and lower limits of each result presented in Tables 4–5 and Figures 2–4 (in previous version Figures 3–5) were calculated with 95% of confidence level. The statistical evaluation methodology was added to 2.3 section. Additionally, correlation between temperature and water absorption as well as swelling in thickness were presented, statistical data was presented in Tables 4–5. However, statistical interpretation of Figure 4 results is not significant because the correlation between constituents is not strong enough. Additionally, bar-type graphs for dimensional stability results are the most common ones.

Point 7: Authors wrote „…Biocomposite boards made of the materials of natural origin are sensible to the environmental factors. In order to increase their longevity and improve the physical-chemical and mechanical properties, impregnation of boards was made using the oils of plant origin...“. These are not conclusions. Authors should delete these sentences.

Response 7: These sentences were deleted. " Still, the sentence "In this work, biocomposite boards were formed and impregnated using the oils of plant origin." was added as an introductory one.

Reviewer 3 Report

The authors presented an interesting study related to physical and mechanical properties of modified biocomposite boards. The paper is easy to read, focuses on the main topic, and brings results that are consistent with other research studies. Results could be used in the processing and better utilization of biocomposite boards.

If flame retardants (i.e. retarders) were not used, it could be said that it comes to a purely organic and ecological product. Biocomposite boards were prepared from hemp shives with the corn starch as a binder and as a hydrophobizer were several types of plant oil. The hydrophilic effects of starch are eliminated by the use of oil, which is beneficial.

The aim was to find the optimal curing temperature for a particular type of oil with respect to selected characteristics. A certain disadvantage is the increase in the thermal conductivity of boards, but a significant improvement in dimensional stability and compressibility is a clear benefit. I consider the total preparation time of such impregnated boards to be a significant disadvantage. The authors should comment on this in a discussion or conclusion, because the application aspect has a significant role here!

I have several formal comments or suggestions for adjustments or additions to the contribution.

Under any graph and table, all the abbreviations used should always be explained, which is not followed in many of them.

In my opinion, there is a lack of information about the pressing cycle in the production of boards, and especially the number and dimensions of boards produced for individual compositions.

At the same time, the methodology lacks information regarding the evaluation of the obtained data. It would be useful to verify the certain significance of your claims, eventually to state that this will require more extensive research. Your test series always has only 3 samples for each property and therefore the statistical significance of the obtained results should be examined, which would be really appropriate to verify.

I consider the research methods used, as well as the introduction to the problematics, to be appropriate and sufficient. I welcome the use of modern research technologies, such as SEM microscopy, and the relevant discussion of the results. However, the reproducibility of the results is difficult to estimate in terms of the number of samples. In conclusion, the authors should comment on the practical application and economic efficiency.

Author Response

Thank you for the valuable remarks on our manuscript. Our responses are presented point-by-point below.

Point 1: I consider the total preparation time of such impregnated boards to be a significant disadvantage. The authors should comment on this in a discussion or conclusion, because the application aspect has a significant role here!

Response 1: The reviewer remark and opinion is really appreciated. Even though the preparation time of BcBs is a really great aspect, however, the environmental friendliness plays a key role for such products. It covers some technological disadvantages which may be optimized. Additionally, being totally ecological, BcBs already have found their application as furniture boards and are being introduced step by step into the market with some additional corrections based on the existing production line. The additional sentence in the Conclusion section about the need to carry out further extensive research in order to optimize technological parameters and preparation duration was added.

Point 2: Under any graph and table, all the abbreviations used should always be explained, which is not followed in many of them.

Response 2: The presence of abbreviations were verified and added where needed (Figures 7 and 9 (current numbering)).

Point 3: In my opinion, there is a lack of information about the pressing cycle in the production of boards, and especially the number and dimensions of boards produced for individual compositions.

Response 3: The pressing cycle is described in section 2.1. "The thermal treatment of biocomposite boards consisted of the following stages: raise of temperature (within 1 h, from the room temperature to 160°C), maintenance of temperature (160°C temperature is maintained for 6 h in the thermal treatment chamber), and the decrease of temperature (within 3 h, till the room temperature)". In total, 6 boards of each composition (6x3) were made and used for the experiments. The dimensions of specimens used for each type of experiments are described in sections 2.1. and 2.3.

Point 4: At the same time, the methodology lacks information regarding the evaluation of the obtained data. It would be useful to verify the certain significance of your claims, eventually to state that this will require more extensive research. Your test series always has only 3 samples for each property and therefore the statistical significance of the obtained results should be examined, which would be really appropriate to verify.

Response 4: Thank you very much for the remark. The methodology used was described in 2.3 section. We do agree that the number of specimens used for each test is rather low, however, statistical evaluation was implemented for each property and upper/lower limits with a 95% confidence interval were presented. Additionally, more samples were tested for water absorption as well as swelling in thickness and the correlations between properties and temperature were presented, statistical data and regression equations were added in Tables 4–5.

Point 5: In conclusion, the authors should comment on the practical application and economic efficiency

Response 5: The practical application of the obtained oil-impregnated biocomposite boards was mentioned in a Conclusion section. These products can be used in furniture industry due to its decorative surface. Additionally, raw materials used for the production of hemp shive biocomposite boards are not hazardous to human health, are environment-friendly and can be an alternative to the existing ones, e.g. wood particle boards that contain formaldehyde or isocyanates.

Round 2

Reviewer 1 Report

Line 104 please correct the word "compsitions"

Lines 106, 108, 109 please use space between the numbers and the units:

it should be 0.924 g/l ± 0.002 g/l instead of 0.924g/l ±0.002g/l

Use the same rule trough the manuscript.

Figure 4 - Please delete the grey horizontal lines inside the plot.

Author Response

Point 1: Line 104 please correct the word "compsitions".

Response 1: Done.

Point 2: Lines 106, 108, 109 please use space between the numbers and the units:

it should be 0.924 g/l ± 0.002 g/l instead of 0.924g/l ±0.002g/l

Use the same rule trough the manuscript.

Response 2: Done in lines 106, 108, 109 and throughout the manuscript. The text was additionally checked for the spelling mistakes and the article usage.

Point 3: Figure 4 - Please delete the grey horizontal lines inside the plot.

Response 3: Done.